# Remdesivir Use in the Real-World Setting: An Overview of Available Evidence

**DOI:** 10.3390/v15051167

**Published:** 2023-05-14

**Authors:** Karolina Akinosoglou, Emmanouil Angelos Rigopoulos, Georgios Schinas, Georgia Kaiafa, Eleni Polyzou, Stamatia Tsoupra, Argyrios Tzouvelekis, Charalambos Gogos, Christos Savopoulos

**Affiliations:** 1Division of Internal Medicine, University General Hospital of Patras, 265 04 Patras, Greece; agrigopoulos@gmail.com (E.A.R.); polyzou.el@gmail.com (E.P.);; 2School of Medicine, University of Patras, 265 04 Patras, Greece; georg.schinas@gmail.com (G.S.); atzouvelekis@upatras.gr (A.T.); cgogos@upatras.gr (C.G.); 31st Medical Propedeutic Department of Internal Medicine, AHEPA, University Hospital of Thessaloniki, Aristotle University of Thessaloniki, 541 24 Thessaloniki, Greece; gdkaiafa@auth.gr (G.K.); csavvopo@auth.gr (C.S.); 4Department of Pulmonology, University General Hospital of Patras, 265 04 Patras, Greece

**Keywords:** remdesivir, Veklury, COVID-19, SARS-CoV-2, real-world evidence, special populations

## Abstract

In the years of Coronavirus Disease 2019 (COVID-19), various treatment options have been utilized. COVID-19 continues to circulate in the global population, and the evolution of the Severe acute respiratory syndrome coronavirus 2 (SARS-CoV-2) virus has posed significant challenges to the treatment and prevention of infection. Remdesivir (RDV), an anti-viral agent with in vitro efficacy against coronaviruses, is a potent and safe treatment as suggested by a plethora of in vitro and in vivo studies and clinical trials. Emerging real-world data have confirmed its effectiveness, and there are currently datasets evaluating its efficacy and safety against SARS-CoV-2 infections in various clinical scenarios, including some that are not in the SmPC recommendations according for COVID-19 pharmacotherapy. Remdesivir increases the chance of recovery, reduces progression to severe disease, lowers mortality rates, and exhibits beneficial post-hospitalization outcomes, especially when used early in the course of the disease. Strong evidence suggests the expansion of remdesivir use in special populations (e.g., pregnancy, immunosuppression, renal impairment, transplantation, elderly and co-medicated patients) where the benefits of treatment outweigh the risk of adverse effects. In this article, we attempt to overview the available real-world data of remdesivir pharmacotherapy. With the unpredictable course of COVID-19, we need to utilize all available knowledge to bridge the gap between clinical research and clinical practice and be sufficiently prepared for the future.

## 1. Introduction

In December 2019 in Wuhan, China, Severe Acute Respiratory Syndrome Coronavirus 2 (SARS-CoV-2) was first identified, calling the world to face an unprecedented health hazard. Due to its rapid transmission, it quickly spread throughout the world, and on 11 March 2020, the World Health Organization (WHO) characterized the outbreak as a pandemic. The COVID-19 pandemic has put a strain on health systems worldwide, causing severe pneumonia and thus increasing hospitalization, intensive care use (ICU) admissions, and mortality rates, especially in people with comorbidities.

SARS-CoV-2 transmission occurs when respiratory droplets and aerosolized viral particles bind to host surface cellular receptors of the upper respiratory tract, conjunctiva, and gastrointestinal tract. The highest risk of transmission occurs in the early phase of the disease, prior to experiencing symptoms. The incubation period is three to five days, depending on protein S variations and virus mutations, while the viral load peaks within one week after symptom onset. Disease progression follows a biphasic course, initially reflecting active viral replication and toxicity, followed by a hyperinflammatory response [1]. Symptomatic COVID-19 varies from mild disease to severe pneumonia, which may lead to hospitalization, respiratory failure, the need for mechanical ventilation, ICU, and death. Multiple vaccines and treatment options have been used to mitigate the effects of COVID-19 on public health. Early intervention, especially in high-risk populations, is pivotal to ensure the best outcomes for patients.

Remdesivir (RDV), branded under the name Veklury, is an antiviral agent that has previously demonstrated antiviral activity against filoviruses (Ebola viruses, Marburg virus), coronaviruses (SARS-CoV, MERS-CoV, SARS-CoV-2), paramyxoviruses (parainfluenza type III virus, Nipah virus, Hendra virus, measles, and mumps virus), and Pnemoviridae (respiratory syncytial virus) [2]. In vitro studies showed that remdesivir exhibited antiviral activity against SARS-CoV-2; thus, it was proposed as an investigational drug early during the pandemic [3]. Consequently, based on data from compassionate use program and clinical trials that demonstrated the superior clinical efficacy of remdesivir to placebo, RDV was first approved by the European Medical Agency (EMA) in July 2020, while the US Food and Drug Administration (FDA) approval followed in October 2020. Initial approval was in the form of conditional marketing authorization, which turned into full marketing authorization in 2022 [4,5].

Real-world evidence (RWE) is an important complement to the Randomized Clinical Trials (RCTs) in a fast-changing pandemic landscape. Constant evolution in disease variants, vaccination status, and populations at risk, as well as approved standards of care, mandate the acquisition of RWE to enable decision making across the diverse spectrum of heterogeneous populations. RWE aids in clarifying therapeutic effectiveness by studying outcomes in heterogenous, more representative patient populations; thus, its role in the drug development process is becoming increasingly important. Due to the rapidly changing nature of the COVID-19 pandemic and the rush to make critical health decisions, multiple real-world studies have been conducted, utilizing information from primary and secondary dataset sources. Primary data typically refer to observational data that are collected in a prospective manner, e.g., cohort and case-control studies. Secondary data consist of electronic health records and hospital chargemaster sources [6]. In this study, we aimed to overview the available evidence of the real-world use of RDV with regards to associated outcomes, including safety and effectiveness, and current trends of use.

## 2. Materials and Methods

Broad searches of Pubmed and peer-reviewed international conferences were conducted using the keywords “Remdesivir” and/or “Veklury” between 1 February 2020 and 20 April 2023. Relevant publications were identified based on the titles and abstracts, and respective references were hand-searched. Clinical trials or experimental data were excluded. Only English language papers were included in this study. Duplicates and irrelevant articles were removed, and all disagreements were discussed and resolved. The study flowchart is shown in Figure 1.

## 3. Remdesivir Outcomes

SARS-CoV-2 is a positive-sense single-stranded RNA virus. RDV is a phosphoramidite prodrug of a monophosphate nucleoside and acts as a viral RNA-dependent RNA polymerase (RdRp) inhibitor, targeting the viral genome replication process [7,8]. Once RDV is metabolized into its pharmacologic active analog adenosine triphosphate (GS-443902), it competes with ATP for integration by the RdRp complex into the nascent RNA strand and, upon subsequent incorporation of a few more nucleotides, results in the termination of RNA synthesis, limiting viral replication. Even higher concentrations of NTP pools can reduce the efficiency of delayed chain termination and result in the formation of full-length RNA products, which retain RDV residues in the primer strand that is later used as a template. Recent data have shown that RDV exhibits inhibitory effects even when present in the template, namely, template-dependent inhibition [9]. RDV’s antiviral activity against SARS-CoV-2 has been previously well demonstrated in vitro and in clinical trials. Potential benefits expand across a wide spectrum of outcomes in the real-world setting, confirming prior experimental data [4].

### 3.1. Clinical Improvement and Increased Chance of Recovery

A number of cohort studies have shown favorable results of RDV treatment with regards to clinical improvement and increased chance of recovery [10,11]. When compared with standard of care in patients with severe COVID-19 and the 14-day clinical recovery determined using a 7-point ordinal scale, RDV was associated with significantly greater recovery (aOR: 2.03 [95% CI: 1.34, 3.08], *p* < 0.001), while mortality was reduced by 62% in patients with severe COVID-19 [11]. Similarly, the WHO 8-point ordinal scale exhibited clinical improvement by day 28 or hospital discharge without worsening of the WHO severity score [12]. In an Italian cohort study, RDV treatment was not associated with a significant reduction of mortality in mechanically ventilated patients, but it was repeatedly associated with a shorter duration of mechanical ventilation (MV) and higher rates of hospital discharge (hazard ratio [HR], 2.25; 95% CI: 1.27–3.97; *p* = 0.005), independent of other risk factors [13]. In the Greek cohort study of 551 patients, a 5-day course of RDV (200 mg on Day 1, 100 mg on Days 2, 3, 4, and 5) given during the first seven days of symptom onset was associated with significantly shorter hospital stay by 2.6 days, lower rates of intubation (*p* = 0.019), higher probability of discharge (*p* = 0.052), and a lower mortality rate (OR 0.38, 95% CI: 0.22–0.67) [14]. These data are in agreement with a retrospective study of five hospitals in the USA comparing the time to clinical improvement versus without RDV. RDV recipients had a shorter time to clinical improvement than controls (median, 5.0 days vs. 7.0 and an adjusted HR of 1.47 [10].

### 3.2. Reduced Disease Progression

Multiple studies showed RDV’s capacity to mitigate the deleterious effects of COVID-19. In a cohort study from Korea, RDV treatment resulted in significantly reduced progression to MV by day 28 (23% vs. 45%; *p* = 0.032), as well as a significantly shorter duration of MV versus supportive care (average 1.97 vs. 5.37 d; *p* = 0.017). Interestingly, rapid viral load reduction was also observed after RDV treatment [15]. When combined with dexamethasone in addition to standard of care, RDV use resulted in a significant reduction in symptom duration, radiographic evidence of pneumonia infiltration, and respiratory failure requiring MV by day 30 (OR 0.36, 95% CI: 0.29–0.46, *p* < 0.0001) compared to standard of care alone in a recent Danish cohort [16]. This comes to no surprise, as the findings of the Adaptive Coronavirus Disease 2019 (COVID-19) Treatment Trial-1 (ACTT-1) suggested that hospitalized COVID-19 patients recovered faster with RDV treatment. A secondary analysis of the data showed higher clinical improvement rates and slower progression to severe disease when treated with RDV. It is indicated that RDV prevents the progression of disease in hospitalized individuals even if they need oxygen supplementation (HR, 0.74; 95% CI: 0.57–0.94) or MV (HR, 0.73; 95% CI: 0.53–1.00) [17].

### 3.3. Reduced Mortality: Treatment Comparison

There is no doubt that RDV reduced the death rate in COVID-19 hospitalized patients with plentiful real-world studies as evidence. A multicenter US study of 24,856 hospitalized patients treated with RDV showed lower inpatient mortality overall (HR 083; 95% CI: 0.79, 0.87) and increased hospital discharge rate by day 28 (HR 1.19; 95% CI: 1.14, 1.25) compared to matched standard of care-treated individuals [18]. Similarly, in another multicenter study of 28,855 individuals, improved survival rates at 14- and 28-day follow-up were concluded in COVID-19 patients treated with RDV (14-day adjusted HR: 0.76; *p* < 0.0001; 28-day adjusted HR: 0.88; *p* = 0.003) [19].

Initially, RDV did not show a mortality benefit in patients without the need of oxygen, contrary to those on low-flow oxygen, where a significant reduction in mortality was achieved (adjusted HR 0.85, 95% CI: 0.77–0.92) [12]. The final results of the Solidarity multinational trial among COVID-19 inpatients confirmed this finding, showing no significant effect of RDV if the patients were already ventilated [20]. However, Mozaffari et al. showed that RDV use was superior even in patients with no need for supplemental oxygen, at 14 and 28 days of follow-up [19]. In agreement with these findings, in non-ventilated hospitalized patients, the likelihood of ventilation and the death rate were reduced [20]. This comes in line with RWE and meta-analysis supporting early administration, as per initial viral replication, and thus the antiviral effect of the regimen [10,21]. Notably, emerging data show that the effect extends across all variant surges [22,23]. The reduction in the mortality rate and hospital admission of outpatient COVID-19-positive individuals has been also compared among nirmatrelvir/ritonavir, molnupiravir, and RDV. Death or hospitalization did not differ among high-risk COVID-19 Italian outpatients treated with currently available antivirals [24].

### 3.4. Benefits of Early Treatment

Clinical experience indicated that delaying RDV treatment could have unfavorable clinical outcomes in terms of higher rates of progression to severe COVID or death in comparison with earlier treated patients. A large observational cohort of 28,855 individuals in the RDV group associated early initiation of treatment with improved survival among patients with COVID-19 [19]. Pre-admission symptom duration was inversely proportionate to the survival rate when treatment began on admission, but the threshold of clinical benefits remained diverse. A retrospective single-center cohort study showed a significantly lower (*p* = 0.001) mortality rate when RDV infusion started within 6 days of symptoms, while there was no significant difference when symptom duration exceeded 6 days [25]. Early administration of a 5-day course of remdesivir (within two days of admission) was associated with a significantly shorter time to clinical improvement, lower viral load [26] and positive IgG antibodies, reduced length of hospital stay, and lower risk of in-hospital death [27]. In another retrospective single-center study, the threshold of clinical benefit was set to 9 days from symptom onset for the initiation of RDV when significantly lower all-cause mortality was documented (*p* = 0.004) [28]. Keeping in mind that testing is delayed for several days following manifestations, other authors showed that if RDV was initiated within three days of a positive test result, the length of stay was shorter (*p* = 0.03), in addition to the mortality rate and the need for MV [29].

In the course of the pandemic, evidence has also been shown regarding RDV as an early outpatient treatment option for COVID-19. In a recent Greek study, 150 individuals were split into two groups that both received RDV treatment. The control group consisted of hospitalized COVID-19 patients with a median of 8 days from symptom onset who were infused RDV for 5 days, while the study group consisted of non-hospitalized COVID-19 positive individuals who received RDV for 3 days and had a median of 4 days from symptom onset. The early 3-day treatment course of RDV prevented progression to critical disease, significantly decreased hospitalizations (*p* < 0.001), respiratory failure (*p* < 0.001), and lowered mortality (*p* = 0.012) [30]. In another study, where early RDV and sotrovimab were used individually as outpatient treatments, hospitalizations and visits to the emergency department were significantly less likely within 29 days from symptom onset (RDV: 11% versus 23.3%; OR = 0.41; sotrovimab: 8% versus 23.3%; OR = 0.28).There was no significant difference between sotrovimab and RDV [31]. Three independent studies (one prospective and two retrospective) later confirmed the significant reduction in hospitalizations and deaths when RDV was initiated early in the progression of COVID-19 [32,33,34].

### 3.5. Reduced Post Hospitalization Outcomes

Even following successful treatment, SARS-CoV-2 sequalae may remain present, affecting patients’ health for months. Post-Acute COVID syndrome (PACS) has been identified, even though the underlying pathogenetic mechanisms remain elusive [35]. Patients require a multisystemic approach and commonly need hospital readmission. RWE has shown that previous RDV treatment showed superior results based on the need for readmission during a 30-day follow-up [36], especially among those with milder disease (RR: 0.31; 95% CI: 0.13, 0.75) [37]. In a real-world cohort analysis of individuals hospitalized with COVID-19 who were admitted to the ICU, those treated with RDV had reduced hospital readmission risk at 30, 60, and 90 days, irrespective of the predominant circulating SARS-CoV-2 variant [38]. RDV treatment exhibits a protective effect against the onset of PACS, showing a 35.9% reduction after 6 months of follow-up (*p* < 0.001) [39]. The results of a long-term follow-up of the randomized SOLIDARITY trial provide no convincing evidence of RDV benefit, but wide confidence intervals included both possible benefit and harm [40]. Further studies are pivotal for safe conclusions to be drawn in this direction.

## 4. Remdesivir in Combination with Other Agents

Numerous studies have examined the effects of combining RDV with other therapeutic agents, especially immunomodulatory regimes in real-world settings.

The combination of RDV with corticosteroids has been dominating the literature. A Chinese cohort study that included 1544 patients evaluated RDV and dexamethasone coadministration in hospitalized patients with moderate-to-severe COVID-19. Patients in the RDV group had a significantly reduced length of hospital stay by 2.65 days (*p* = 0.002), lower WHO clinical progression scale scores in the 30-, 60- and 90-day follow-up periods (*p* < 0.001), and reduced in-hospital mortality (*p* = 0.042) [41]. A similar study in Italy showed a significantly reduced death rate (*p* < 0.003) and length of hospital stay (*p* < 0.0001) in the RDV combined with dexamethasone group compared to dexamethasone alone. Rapid elimination of SARS-CoV-2 was observed (median 6 vs. 16 days; *p* < 0.001) [41]. Among COVID-19 patients with complicated disease or those admitted to the ICU, those with concurrent use of corticosteroids displayed a reduced in-hospital mortality rate [42,43], in agreement with previous results from the RECOVERY trial [44]. Treatment with dexamethasone, RDV, or both in patients hospitalized with COVID-19 was associated with a lower frequency of neurological complications in an additive manner, such that the greatest benefit was observed in patients who received both drugs together [45].

The combination of RDV with the immunomodulating agents tocilizumab/baricitinib achieved shorter respiratory recovery time (median 11 versus 21 days, *p* = 0.033) and better respiratory status at the 14- and 28-day follow-up, as shown in a retrospective Japanese study [46]. Experience from Spain came to show that the combination of RDV with dexamethasone and tocilizumab decreased the risk of all-cause 28-day mortality and the need for invasive MV in patients with high viral loads and low-grade systemic inflammation [47].

Current trends examine the combination of RDV with other antivirals or monoclonal agents, especially in the context of immunosuppression and persistent disease, as well as breakthrough or re-infection. With clinical trial data under way, RWE is useful (Table 1) (See next section for details).

## 5. Special Populations

Patients with multiple co-morbidities and special conditions were at particular high risk for adverse outcomes following infection with SARS-CoV-2. Post vaccination statistics showed lower induced immune responses in immunocompromised patients, especially those with hematological malignancies and solid organ transplant recipients [59]. Although patients with common co-morbidities e.g., hypertension, pulmonary disease etc., are included in many clinical trials, special populations, e.g., pregnant women, patients on renal replacement, and hematologic patients, are often underrepresented due to bioethics, legislation or vulnerability. Evidence mostly comes a posteriori following implementation in clinical practice.

RDV has shown nephrotoxicity in non-human studies. Even though the respective toxic doses were 3.5 times higher than those used in treatment, RDV use in individuals with impaired renal function is a safety concern. A high number of studies assessed the efficacy and safety of RDV in patients with chronic kidney disease, patients on dialysis, and kidney transplant recipients [61,62,63,64,65,66,67,68,69,70]. RDV was well tolerated in patients with acute kidney injury or chronic kidney disease [63] while reducing the risk of mortality in patients on dialysis [64]. RDV treatment within 48 h shortened the time to recovery and discharge in patients with end-stage renal disease [66]. All real-world studies showed that RDV was relatively safe and well tolerated in patients with severe renal disease [62,63,65,81]. The results of RWE and emerging needs also drove the Phase 3, randomized, REDPINE study, recruiting patients with eGFR < 30 mL/min/1.73 m^2^, regardless of the need for dialysis. Even in this population, RDV was safe and well tolerated, while no dose adjustment was required [82].

Pregnant women when compared to similar age non-pregnant individuals are more likely to require hospitalization or even admission to the ICU and need invasive MV [71]. They have been historically excluded from drug development research protocols in fear of adverse pregnancy and maternal health outcomes [83]. RWE from the US showed that RDV was a potent and low-risk treatment option, promoting clinical improvement, lowering ICU and death rates, and resulting in a low incidence of serious adverse effects [71,72,73,74].

The evidence of RDV efficacy and safety among immunocompromised populations has come from cohorts or case series, highlighting the potential of a combination of antiviral therapies or prolonged courses of RDV (Table 1) with variable results [32,48,49,50,51,52,53,54,55,56,57,58,59,60,61,62,63,64,65,66,67,68,69,70,71,72,73,74,75,76,77,78,79,80]. Persistent infection in immunocompromised patients remains a concern, as it seems to drive virus evolution and the selection of new variants [84]; hence, experience of individualized successful treatment is necessary. Nonetheless, recent data have shown that RDV-associated reduction in mortality in hospitalized cancer patients was consistently observed across all variants of concern prior to B4/5 [85], while its use reduced re-admission in immunocompromised patients [86]. The available RWE in special populations is summarized in Table 1, including patients with chronic liver disease [77,87], those prone to liver damage [79], the pediatric population [75,76,88], and very elderly [78].

## 6. Resistance/Mutations

RDV has been recognized as a broad-spectrum antiviral agent, effectively inhibiting the replication of SARS-CoV-2 by binding to the viral RNA-dependent RNA polymerase (RdRp) [89]. Its unique capacity as an analog allows it to evade the proofreading activity of the viral exoribonuclease, thereby conferring its efficacy against all known coronaviruses [90]. However, after more than 2 years of widespread RDV use during the COVID-19 pandemic, concerns have arisen regarding the potential development of SARS-CoV-2 resistance mechanisms. Surveillance efforts to identify resistance-associated mutations have been limited, as the specific resistance patterns of SARS-CoV-2 remain largely unknown due to its insidious onset and sudden emergence.

Prior to the COVID-19 pandemic, mutations in the conserved nsp12 protein of RdRp, found across all coronaviruses, had been proposed and confirmed to confer resistance against RDV [90]. Consequently, research efforts have primarily focused on this region of the viral genome [91]. A recent exploratory analysis of SARS-CoV-2 genomes revealed low and stable levels of RDV resistance in a real-world setting, with circulating mutated variants conferring resistance to RDV being associated with poor viral fitness [92]. Until recently, only two amino acid substitutions, D484Y and E802D, had been identified in vivo in immunocompromised COVID-19 patients who received intravenous RDV treatment [93,94]. Commonalities between the two cases include prolonged viral shedding due to the patients’ immunocompromised status and underlying lung involvement, raising questions about compartmentalized resistance given RDV poor lung penetration [95,96].

A study investigating the diversity of SARS-CoV-2 within 14 individual patients over time and the impact of antiviral treatment on the virus found that RDV treatment could rapidly fix newly acquired mutations, adding to the list of concerns [97]. Moreover, a pre-print case series of renal transplant recipients reports the development of a newly discovered V792I mutation in SARS-CoV-2’s RdRp following RDV treatment, further emphasizing the need for careful isolation precautions in immunocompromised hosts to prevent the spread of mutated SARS-CoV-2 isolates [98].

## 7. Limitations

A number of limitations are identified in this review. Even though a systematic and transparent description of methods is presented in the respective sections and Figure 1, bias in the selection and interpretation of studies is possible, as in every literature review. Moreover, we performed a literature review in a rapidly evolving field that has utterly changed since the beginning of the pandemic. Comparison studies included did not involve regimens that at the moment are no longer recommended, e.g., lopinavir/ritonavir and hydroxychloroquine, as no true power of evidence would be added. In this context, it is possible that by the time this manuscript is published, RDV indications may be outdated or more data will be available. In addition, only English-language papers were reviewed. Experience from RDV recorded in non-English literature contributing significant input may have been missed. Moreover, one cannot overcome potential publication bias that already excludes negative results. Grey or unpublished literature was not assessed in this report.

Last, the authors have to recognize potential random and systematic errors that have to be accounted for in real-world study populations before safe conclusions are drawn. Even though random errors caused by population heterogenicity can be addressed with confidence intervals when the sample size is sufficient, systematic errors (bias) including immortal time bias and cofounders, should be taken into account when RWE is assessed. The implementation of methodological approaches such as propensity score matching, risk set sampling, and multivariate analysis helps balance heterogenic features of populations. It is of great importance to equally understand the methods used to achieve sufficient quality of evidence.

## 8. Expert Opinion and Future Directions

An evidence gap between clinical practice and clinical research calls for harnessing real-world data [99]. RDV (Veklury^®^) has been proven safe and effective in the treatment of COVID-19 in hospitalized patients, evident by multiple and variable real-world studies. Experience has come to add up from the beginning of the pandemic and extends across a diverse spectrum of disease (early disease with risk factors not requiring oxygen to severe disease requiring oxygen), populations, and/or co-morbidities. The benefits of timely treatment and studies showing similar clinical outcomes in outpatients underline the need for early COVID-19 diagnosis while expanding access to outpatient antimicrobial treatment facilities. Several studies showed that RDV infusion in the outpatient setting seems to be a safe and efficient alternative to conventional hospitalization for treating non-severe COVID-19 patients [100]. In one of the studies, only 2% reported adverse effects and only 5% required hospitalization [101].

Following extensive vaccination coverage, a shift in the affected population has been observed. At the moment, COVID-19 tends to dominate the elderly and/or severely immunocompromised high-risk populations, commonly already hospitalized for reasons other than COVID-19. In the setting of polypharmacy and a lack of drug–drug interactions, and importantly, the absence of specific or strict eligibility criteria as occurs with other regimens, RDV is an ideal candidate for high-risk individuals, asymptomatic early in the course of disease, with an equally safe profile providing a “low burden” of once-a-day infusion. In this case, compliance is ensured as per RDV’s IV administration that requires the presence of a health care provider.

However, the issue of persistent or breakthrough infections in immunocompromised populations remains of concern. Simultaneous or sequential combination with other antivirals and/or repeated or prolonged courses have been proposed. Different mechanisms of action across various available antivirals can potentiate synergistic effects when co-administered, with possibly faster and more effective viral clearance in this population [54,57,60]. In view of emerging RWE suggesting that when used for extended or repeated treatment courses, RDV leads to improved clinical and laboratory findings and eventually discharge, large clinical trials are now in progress [48,49,53,56,60] (Table 1).

At present, it is not possible to definitively state that repeated administration of RDV will pose significant challenges in clinical settings. The identified mutations associated with resistance have been found to compromise viral fitness, suggesting that the therapeutic value of RDV is still largely intact [91,92]. A recent in vitro study revealed that introducing the V166L mutation—the sole Nsp12 substitution after 17 passages under RDV, located outside the polymerase active site—into a recombinant SARS-CoV-2 virus led to a modest 1.5-fold increase in EC50, indicating a quite high in vitro barrier to RDV resistance [102]. This comes in line with SARS-CoV-2 resistance analyses from the Phase 3 PINETREE trial indicating a high barrier to the development of RDV resistance in COVID-19 patients [103]. Nevertheless, as our understanding of SARS-CoV-2 resistance mechanisms advances, the pursuit of novel antivirals exhibiting greater affinity for the RdRp complex may become increasingly important.

Such compounds would not only enhance the efficacy of current antiviral agents but also help safeguard against potential future resistance developments. Ideally, a promising future oral version of the RDV compound would exhibit enhanced oral bioavailability and plasma half-life, allowing for significantly and quickly reduced viral loads, tissue-specific localization, and decreased lung injury. Adequate safety and resistance profiles would be ideal, so that maximally maintained antiviral activity against different SARS-CoV-2 variants is ensured.

RDV has been the intravenously administered version (version 1.0) of GS-441524. In that sense, oral GS-441524 derivatives (VV116 and ATV006; version 2.0, targeting highly conserved viral RdRp) could be game-changers in treating COVID-19, as oral administration has the potential to maximize clinical benefits, including a decreased duration of COVID-19 and a reduced post-acute sequelae of SARS-CoV-2 infection, as well as limited side effects such as hepatic accumulation [104]. Nonetheless, the gap between bench and bedside is not to be minimized soon if (1) large-scale manufacturing ability for increased accessibility and affordability to outpatients is not ensured and (2) RWE for better understanding of the clinical phenotype diversity and impact of intervention is not efficiently collected.

## Figures and Tables

**Figure 1 viruses-15-01167-f001:**
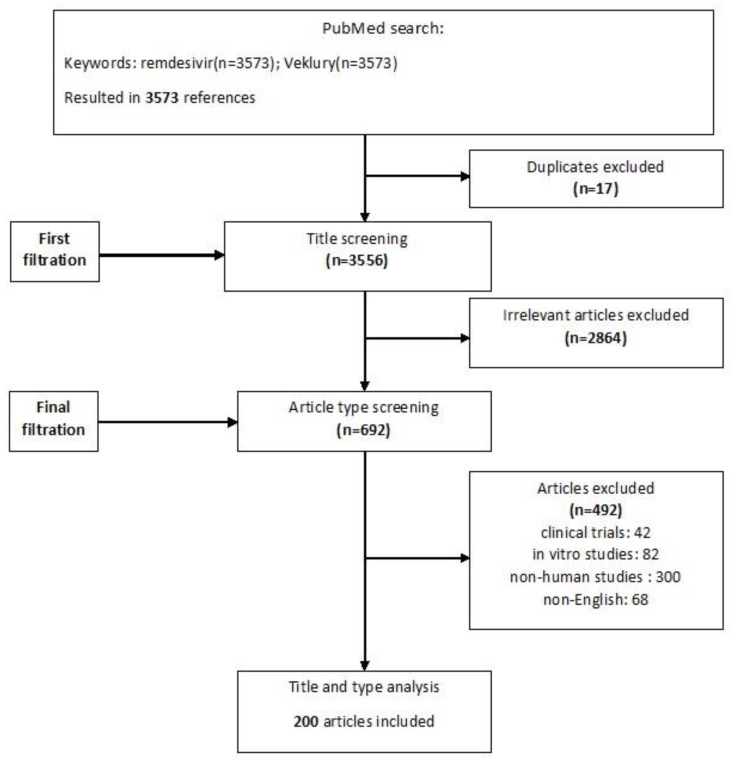
Flowchart of included and excluded studies.

**Table 1 viruses-15-01167-t001:** Remdesivir experience in special populations.

Reference	Country of Origin	Population Characteristics	Number of Patients	Therapeutic Care	Endpoints
Immunocompromised individuals					
Ford, E. S., et al. (2022). Successful treatment of prolonged, severe COVID-19 lower respiratory tract disease in a B-cell ALL patient with an extended course of remdesivir and nirmatrelvir/ritonavir. Clin Infect Dis. [48] https://doi.org/10.1093/cid/ciac868	USA	40-year-old man B-cell acute lymphoblastic leukemia with complete remission Under chemotherapy Prophylaxis with Casirivimab and Imdevimab prolonged, persistent, and severe lower respiratory tract COVID-19	1	Multiple therapies tried 10 days IV remdesivir, dexamethasone, and nirmatrelvir/ritonavir 20 days nirmatrelvir/ritonavir	Virus remained detectable by BAL, and oxygen requirement returned after 10 days of nirmatrelvir/ritonavir in combination with remdesivir Sustained improvement and clinical cure: 20 days of nirmatrelvir/ritonavir
Dioverti, M. V., et al. (2022). Combination Therapy With Casirivimab/Imdevimab and Remdesivir for Protracted SARS-CoV-2 Infection in B-cell-Depleted Patients. Open forum infectious diseases, 9(6), ofac064. [49] https://doi.org/10.1093/ofid/ofac064	USA	B-cell-depleted patients	3	Patient 1: Casirivimab/Imdevimab on day 70 Remdesivir on days 75–79 Patient 2: Casirivimab/Imdevimab on day 111 Remdesivir on days 115–119 Patient 3: Casirivimab/Imdevimab on day 47 Remdesivir on days 46–50	Patient 1: SARS-CoV-2 test negative on day 84 Resolution of cough by 2 weeks Normalization of chest CT by 4 weeks Leukopenia/ neutropenia normalized by 2 weeks Clinically well 10 months after treatment Patient 2: Negative test on day 146 Clinically well 10 months after treatment Patient 3: Negative test on day 64 Clinically well 8 months after treatment
Rajakumar, I., et al. (2022). Extensive environmental contamination and prolonged severe acute respiratory coronavirus-2 (SARS CoV-2) viability in immunosuppressed recent heart transplant recipients with clinical and virologic benefit with remdesivir. Infection Control & Hospital Epidemiology, 43(6), 817–819. [50] https://doi.org/10.1017/ice.2021.89	UK	Immunosuppressed recent heart transplant recipients with benefit with remdesivir	2	Case 1: 10 day course of remdesivir on day 23 Case 2: 5 day course of remdesivir on day 16	Case 1: clinical condition and chest radiograph improved by day 44 Case 2: Clinical recovery and discharge on day 23
Baldi, F., et al. (2023). Case report: Sotrovimab, remdesivir and nirmatrelvir/ritonavir combination as salvage treatment option in two immunocompromised patients hospitalized for COVID-19. Frontiers in medicine, 9, 1062450. [51] https://doi.org/10.3389/fmed.2022.1062450	Italy	Immunocompromised individuals treated with rituximab in need of hospitalization for COVID-19	2 Case1: non-Hodgkin lymphoma Case 2: granulomatosis with polyangiitis	A single infusion of IV sotrovimab 500 mg, a 7-day course of IV remdesivir (200 mg of loading dose, 100 mg of maintenance dose) plus 5 days of oral nirmatrelvir/ritonavir 300 mg/100 mg q12h. Case 1: On day 7: initiation of treatment plus IV corticosteroids Case 2: On day 6: initiation of treatment	Case 1: clinical improvement on day 9, no need for oxygen supplementation by day 13. Discharged on day 17 Case 2: clinical improvement on day 11, no need for oxygen supplementation by day 12. Discharged on day 14
Trottier, C. A., et al. (2023). Dual Antiviral Therapy for Persistent Coronavirus Disease 2019 and Associated Organizing Pneumonia in an Immunocompromised Host. Clinical infectious diseases: an official publication of the Infectious Diseases Society of America, 76(5), 923–925. [52] https://doi.org/10.1093/cid/ciac847	USA	Immunocompromised patient with chronic lymphocytic leukemia with persistent COVID-19	1	First admission: 10 days of remdesivir plus iv corticosteroids. Second admission: received bebtelovimab Third admission: IV corticosteroids with remdesivir and nirmatrelvir/ritonavir for 20 days	First admission: Clinical improvement, relapse in one month. Readmitted two months later Second admission: Clinical improvement, relapse in one month and readmission Third admission: Clinical improvement by day 3 By day 9, discharged No relapse and no adverse events two months later.
Buckland, M. S., et al. (2020). Treatment of COVID-19 with remdesivir in the absence of humoral immunity: a case report. Nature communications, 11(1), 6385. [53] https://doi.org/10.1038/s41467-020-19761-2	UK	Patient with genetic antibody deficiency X-linked agammaglobulinemia (XLA) with persistent COVID-19	1	10 days of remdesivir on day 34 Readmitted on day 54 10 more days of remdesivir on day 61	First admission: Fever, dyspnea improved by 36 h Nausea, vomiting ceased Oxygen saturation raised CRP decreased Rise in lymphocyte count Improved CT findings Discharged on day 43 Second admission: Similar results Discharged on day 73
Helleberg, M., et al. (2020). Persistent COVID-19 in an Immunocompromised Patient Temporarily Responsive to Two Courses of Remdesivir Therapy. The Journal of infectious diseases, 222(7), 1103–1107. [54] https://doi.org/10.1093/infdis/jiaa446	Denmark	Immunocompromised patient due to treatment for chronic lymphocytic leukemia with persistent COVID-19 with high fever and severe pneumonia	1	Two 10-day courses of remdesivir (200 mg first dose and 100 mg per day after) On day 24 and day 45	First treatment course (days 26–35): On day 26: Fever abated General condition improved Discharged on day 35 Relapsed and readmitted on day 36 Second treatment course (days 45–54) Afebrile on day 46 Discharged on day 54 with negative nasal swab On day 55: Fever recurred Readmitted with positive test On day 58, the patient received convalescent plasma Discharged on day 65
Biscarini, S., et al. (2022). Safety Profile and Outcomes of Early COVID-19 Treatments in Immunocompromised Patients: A Single-Centre Cohort Study. Biomedicines, 10(8), 2002. [32] https://doi.org/10.3390/biomedicines10082002	Italy	Non-hospitalized people who received early treatment with remdesivir or monoclonal antibodies	143 106/143 (74.1%) immunocompromised (41 solid organ transplant recipients, 6 hematopoietic stem cell transplant recipients) 37/143 not immunocompromised	23 treated with remdesivir 122 treated with monoclonal antibodies 2 treated with both	There were no statistically significant differences between the two groups in AEs (23/143 or 16.1%) Mean duration of symptoms after treatment onset: 2.5 days (IQR: 1.0–6.0) molecular swab positivity: 10 days (IQR: 6–16) Clinical outcome: hospital or ICU admission: 5/143 (3.5%) mortality: 1/143 (0.7%)
Martinez, M. A., et al. (2022). Extended Remdesivir Infusion for Persistent Coronavirus Disease 2019 Infection. Open Forum Infectious Diseases, 9(8), ofac382. [55] https://doi.org/10.1093/ofid/ofac382	USA	Immunocompromised patient with granulomatosis with polyangiitis and secondary hypogammaglobulinemia treated with multiple immunocompromising agents	1	First admission: remdesivir IV for 5 days plus corticosteroids IV for 10 days Second admission on day 18: Convalescent plasma Third admission on day 52: Only antibiotic therapy Fourth admission on day 75: remdesivir and IV corticosteroids on day 94 for 10 days Fifth admission on day 110: convalescent plasma and remdesivir for 30 days	After 30 days of remdesivir treatment on the fifth admission, the patient was discharged on day 142. At 12-month follow-up: No major adverse events reported No relapse at 12-month follow-up
Camprubí, D., et al. (2021). Persistent replication of SARS-CoV-2 in a severely immunocompromised patient treated with several courses of remdesivir. International journal of infectious diseases: IJID: official publication of the International Society for Infectious Diseases, 104, 379–381. [56] https://doi.org/10.1016/j.ijid.2020.12.050	Spain	Severely immunocompromised patient in treatment for stage IV-A follicular lymphoma	1	200 mg first dose and 100 mg per day for 8 days On day 28, a single dose of remdesivir was accidentally administered 10-day course of remdesivir at second admission	First treatment course (days 28–36): Discharged on day 36 Relapsed and readmitted on day 39 Afebrile 1 day after the accidental remdesivir admission Second treatment course: Rapid clinical improvement Fever resolved Short courses of remdesivir might be insufficient for treating high-risk populations
Fesu, D., et al. (2022). Remdesivir in Solid Organ Recipients for COVID-19 Pneumonia. Transplantation proceedings, 54(9), 2567–2569. [57] https://doi.org/10.1016/j.transproceed.2022.10.043	Hungary	Solid organ transplant recipients Mean age 53.2 ± 12.7 Control group: non-transplanted patients who received remdesivir (RDV group) and standard of care (SOC group)	25 transplant recipients (lung: 19; kidney: 3; liver: 2; heart: 1)	15/25 treated with remdesivir (RDV-TX group) 10/25 received standard of care treatment (SOC-TX)	Safety and efficacy of remdesivir was assessed: worse clinical score was noted in RDV patients compared with RDV-TX transfer to ICU was worse in RDV-TX group compared to RDV 60-day survival was worse in RDV-TX group compared to RDV All 60-day fatalities were among the lung transplant recipients (6/19) No AEs were noted related to remdesivir therapy Remdesivir use was safe in solid organ transplant recipients; however, outcome was significantly worse compared to the non-transplanted individuals
Colaneri, M., et al. (2022). Early remdesivir to prevent severe COVID-19 in recipients of solid organ transplant: a real-life study from Northern Italy. International journal of infectious diseases: IJID: official publication of the International Society for Infectious Diseases, 121, 157–160. [58] https://doi.org/10.1016/j.ijid.2022.05.001	Italy	Outpatient solid organ transplant recipients who received and did not receive pre-emptive remdesivir were evaluated	24 7/24 (29.1%) treated with pre-emptive remdesivir	3-day course of remdesivir to prevent hospitalization.	Remdesivir prevented hospitalization at day 28 after the positive test: Hospitalization or COVID-19 worsening: remdesivir group 0/7 vs. non-remdesivir group 9/17 (1/9 ICU admission) (aHR 0.05; CI [0.00–0.65], *p*-value = 0.01) Time to negative SARS-CoV-2 RT-PCR swab test result was shortened No differences between the two groups was found (*p*-value = 0.86).
Lafont, E., et al. (2022). Targeted SARS-CoV-2 treatment is associated with decreased mortality in immunocompromised patients with COVID-19. The Journal of antimicrobial chemotherapy, 77(10), 2688–2692. [59] https://doi.org/10.1093/jac/dkac253	France	Immunocompromised patients with COVID-19	67	10 did not receive targeted treatment 22 remdesivir 16 sotrovimab 13 tixagevimab/cilgavimab 1 Casirivimab/Imdevimab	Mortality after targeted treatment vs. no targeted treatment and safety of use: Mortality: Significantly lower in treatment patients [n = 0 (0%) vs. n = 2 (20%); *p* = 0.034] Safety: No severe adverse events were reported among treated patients Targeted COVID-19 treatment is safe and efficient and could be proposed in high-risk immunocompromised patients.
Shields, A. M., et al. (2022). Outcomes following SARS-CoV-2 infection in patients with primary and secondary immunodeficiency in the UK. Clinical and experimental immunology, 209(3), 247–258. [60] https://doi.org/10.1093/cei/uxac008	UK	Patients with primary (PID) and secondary immunodeficiency (SID) and COVID-19	310	33 received targeted treatment 26/33 received remdesivir	Mortality rate 55/310 (17.7%) CVID: Infection fatality rate (IFR): 17/93 (18.3%) PID: IFR: 26/159 (16.3%) SID: IFR: 25/92 (27.2%)
Patients with impaired renal function					
Lim, J. H., et al. (2022). “Clinical Effectiveness and Safety of Remdesivir in Hemodialysis Patients with COVID-19.” Kidney Int Rep 7(11): 2522–2525. [61] https://www.ncbi.nlm.nih.gov/pubmed/36105653	South Korea	Mean age: 68.5 ± 12.8 years 66.1% male hospitalized patients with COVID-19 who are on hemodialysis	118 in total 44 received remdesivir 74 did not receive remdesivir	Remdesivir at half the standard dose Loading dose 100 mg Maintenance dose was 50 mg for the next 2 to 4 days	The remdesivir group had a tendency of more severe disease (*p* = 0.058) The NEWS on the day of hospitalization was significantly higher in the remdesivir group (*p* = 0.026) Mortality: 1/44 [2.3%] vs. 5/74 [6.8%] (*p* = 0.284) The composite outcome of mortality, use of a high-flow nasal cannula, and transfer to the intensive care unit occurred less frequently in the remdesivir group (1 [2.3%] vs. 10 [13.5%], *p* = 0.042) Disease severity aggravation rate was lower in the remdesivir group (3 [6.8%] vs. 15 [20.3%], *p* = 0.049)
Ackley, T. W., et al. (2021). A Valid Warning or Clinical Lore: an Evaluation of Safety Outcomes of Remdesivir in Patients with Impaired Renal Function from a Multicenter Matched Cohort. Antimicrobial agents and chemotherapy, 65(2), e02290-20. [62] https://doi.org/10.1128/AAC.02290-20	USA	Hospitalized patients with SARS-CoV-2 with impaired renal function Group 1: patients with an estimated creatinine clearance (eCrCl) of <30 mL/min (median, 80 years [64 to 89]) Group 2: eCrCl of ≥30 mL/min (median, 62 years [54 to 74])	359 in total 347 included N (Group 1) = 40 N (Group 2) = 307	Days of remdesivir: Group 1: median 5 [IQR 4–5] Group 2: median 5 [IQR 5–5]	No significant difference in the frequency of end of treatment AKI (5% versus 2.3%; *p* = 0.283) or early discontinuation due to abnormal liver function tests (LFTs) (0% versus 3.9%; *p* = 0.374) Mortality rate was higher in group 1 (50% versus 16.2%; *p* < 0.001)
Thakare, S., et al. (2021). Safety of Remdesivir in Patients With Acute Kidney Injury or CKD. Kidney international reports, 6(1), 206–210. [63] https://doi.org/10.1016/j.ekir.2020.10.005	India	Patients With acute kidney injury or CKD	157 in total, 46 received remdesivir, 8/46 were kidney transplant recipients	Remdesivir was administered as a total dose of 600 mg (200 mg on day 1, followed by 100 mg/day) 2 patients received 1200 mg because satisfactory clinical improvement was not observed	AST/ALT levels: baseline test abnormalities in 14/46 (30.4%) 12/14 improved by the end of the therapy Stable test results in 28/46 (60.9%) 3/46 (6.5%) newly occurring test abnormalities during therapy Clinical outcomes: 14/46 (30.4%) died 24/46 (52.2%) recovered 8/46 (17.3%) still admitted No renal function abnormalities attributable to the drug were observed
Kikuchi, K., et al. (2021). Survival and predictive factors in dialysis patients with COVID-19 in Japan: a nationwide cohort study. Renal replacement therapy, 7(1), 59. [64] https://doi.org/10.1186/s41100-021-00378-0	Japan	Patients on dialysis with COVID-19	1948 in total 1010 included 392 analyzed for the efficacy of remdesivir	98/392 received remdesivir 294/392 did not receive remdesivir	Mortality risk: increased with age (*p* < 0.001). significantly higher in patients with peripheral arterial disease (HR: 1.49, 95% CI: 1.05–2.10) significantly lower in patients who were treated with remdesivir (HR: 0.60, 95% CI: 0.37–0.98) increased with increase in BMI, increased with increase in CRP, decreased with increase in albumin. Length of stay: 20.9 ± 13.2 days in the remdesivir group, 16.2 ± 8.1 days in the other group (Difference: 4.7 days, 95% CI: 2.2–7.4, *p* < 0.001). Clinical outcomes 699/1010 (69.2%) recovered 311/1010 (30.8%) died
Pettit, N. N., et al. (2021). Remdesivir Use in the Setting of Severe Renal Impairment: A Theoretical Concern or Real Risk?. Clinical infectious diseases: an official publication of the Infectious Diseases Society of America, 73(11), e3990–e3995. [65] https://doi.org/10.1093/cid/ciaa1851	USA	Hospitalized patients (median age: 50) and patients with severe renal impairment (SRI, creatinine clearance < 30 mL/min or requiring renal replacement therapy) (median age: 74)	137 in total 135 included N(SRI) = 20	5-day course of remdesivir	Safety assessment of remdesivir in patients with SRI: The incidence of possible AEs was 30% among those with SRI vs. 11% without (*p* = 0.06). Liver function test (LFT) elevations occurred in 10% vs. 4% (*p* = 0.28) Serum creatinine (SCr) elevations in 27% vs. 6% (*p* = 0.02) Mortality and length of stay were consistent with historical controls. The use of remdesivir in this small series of patients with SRI appeared to be relatively safe
Aiswarya, D., et al. (2021). Use of Remdesivir in Patients With COVID-19 on Hemodialysis: A Study of Safety and Tolerance. Kidney international reports, 6(3), 586–593. [66] https://doi.org/10.1016/j.ekir.2020.12.003	India	Dialysis-dependent patients with COVID-19 Mean age 50.1 ± 12.2 years	48 38 men 10 women	100 mg of remdesivir 4 h before dialysis Max. of 6 doses	Before and after treatment comparison: Liver function: No events of significant alterations CRP: significant decline was noted (*p* = 0.001) O2 requirements: 68.6% improved Length of stay: shortened by mean 5.5 days (*p* = 0.001)
Elec, F., et al. (2022). COVID-19 and kidney transplantation: the impact of remdesivir on renal function and outcome—a retrospective cohort study. International journal of infectious diseases: IJID: official publication of the International Society for Infectious Diseases, 118, 247–253. [67] https://doi.org/10.1016/j.ijid.2022.03.015	Romania	Kidney transplant recipients	165	200 mg of remdesivir on day 1 followed by 100 mg for 2 to 10 days 5-day course for 38/165 Standard of care for 127/165	Impact of remdesivir on: Overall mortality: no difference (18% vs. 23%, *p* > 0.05) ICU mortality: significant reduction (39% vs. 83%, *p* < 0.05) Incidence of AKI: no difference (50% vs. 43%, *p* > 0.05) Discharge eGFR values: significant improvement (57 ± 23 vs. 44 ± 22, *p* < 0.05)
Butt B, et al. Efficacy and Safety of Remdesivir in COVID-19 Positive Dialysis Patients. Antibiotics. 2022; 11(2):156. [68] https://doi.org/10.3390/antibiotics11020156	Pakistan Saudi Arabia	Patients in dialysis with COVID-19	83 51/83 received remdesivir within 2 days of symptom onset 32/83 received remdesivir more than 2 days after symptom onset	100 mg of remdesivir before hemodialysis	Mortality rate: Remdesivir did not reduce the overall mortality (76/83 or 91.5% survived while 7/83 or 8.4% died) Serum CRP (*p* = 0.028) and total leucocyte count (*p* = 0.013) increased 30-day mortality Compared remdesivir administration within 2 days versus after more than 2 days: Length of hospitalization was lower (*p* = 0.03) in the first group Nasal swabs were negative earlier (*p* = 0.001) in the first group Adverse events: No major AEs observed
Estiverne, C., et al. (2021). Remdesivir in Patients With Estimated GFR < 30 mL/min per 1.73 m^2^ or on Renal Replacement Therapy. Kidney international reports, 6(3), 835–838. [69] https://doi.org/10.1016/j.ekir.2020.11.025	USA	COVID-19 positive patients with eGFR < 30 mL/min or on renal replacement therapy Median age 63 (IQR 55–76)	18 13 with low eGFR 5/13 AKI 8/13 stable chronic kidney disease 5 received renal replacement therapy 11 already in ICU 9 already in need of mechanical ventilation	200 mg first dose and then 100 mg per day	AEs after remdesivir treatment: Changes in ALT: 2 patients developed >5 times ALT levels not attributed to remdesivir Changes in serum creatinine: 8/13 improved creatinine levels 5/13 worsened creatinine levels (in 1 case attributed to remdesivir Overall mortality: 8/18 on day 28 All non-ICU patients survived
Stancampiano, F., et al. (2022). Use of remdesivir for COVID-19 pneumonia in patients with advanced kidney disease: A retrospective multicenter study. Clinical infection in practice, 16, 100207. [70] https://doi.org/10.1016/j.clinpr.2022.100207	USA	Patients with advanced kidney disease (eGFR < 30) Median age 72 (21–100)	444 114 (25.7%) with stage 3 CKD 229 with stage 4 CKD (51.6%) 101 with stage 5 CKD (22.7%)	200 mg first dose and 100 mg per day for 5 days total	Clinical outcomes: ICU admissions: 146/444 (32.9%) Deaths 103/444 (23.2%) Patients in need of dialysis during treatment: 120/444 (27%) AEs did not differ between stage groups (*p* = 0.12) Stage 3 CKD: 20.9% Stage 4 CKD: 30.2% Stage 5 CKD: 32.3% Suggests that the use of remdesivir is safe in patients with severe CKD
Pregnant women					
Burwick, R. M., et al. (2021). Compassionate Use of Remdesivir in Pregnant Women With Severe Coronavirus Disease 2019. Clinical infectious diseases: an official publication of the Infectious Diseases Society of America, 73(11), e3996–e4004. [71] https://doi.org/10.1093/cid/ciaa1466	USA	Pregnant women with severe COVID-19 (saturation ≤ 94%)	86 19/86 delivered before the initiation of remdesivir (postpartum group) 67/86 in pregnant group	200 mg of remdesivir on day 1 followed by 100 mg for 2 to 10 days	Invasive ventilation required: 40% of pregnant women 95% of postpartum women At 28-day follow-up, oxygen requirement decreased in 96% of pregnant women 89% of postpartum Among women on mechanical ventilation: Pregnant group 93% extubated 93% recovered 90% discharged Postpartum group 89% extubated 89% recovered 84% discharged There was 1 maternal death attributed to neonatal disease There were no neonatal deaths recorded There was low incidence of serious AEs (16%)
Eid, J., et al. (2022). Early Administration of Remdesivir and Intensive Care Unit Admission in Hospitalized Pregnant Individuals With Coronavirus Disease 2019 (COVID-19). Obstetrics and gynecology, 139(4), 619–621. [72] https://doi.org/10.1097/AOG.0000000000004734	USA	Hospitalized pregnant women with COVID-19 All of them unvaccinated	41 Divided into groups by day of initiation of remdesivir Early group (<7 days after the onset of symptoms) N = 24 Late group (≥7 days after the onset of symptoms) N = 17	Early group: mean duration of therapy: 3 days (1–6 days) Late group: mean duration of therapy: 9 days (7–14 days)	Comparison between early and late groups showed Lower ICU admission rate: 21% vs. 59% (OR: 0.18 [95% CI: 0.04–0.72]) Lower length of hospital stay: 5 [IQR 4–5.75] vs. 11 [IQR 4.5–15.5] *p* < 0.01 Decreased progression to critical disease: 12% vs. 4% (OR: 0.20 [95% CI: 0.51–0.87])
Nasrallah, S., et al. (2022). Pharmacological treatment in pregnant women with moderate symptoms of coronavirus disease 2019 (COVID-19) pneumonia. The journal of maternal-fetal & neonatal medicine: the official journal of the European Association of Perinatal Medicine, the Federation of Asia and Oceania Perinatal Societies, the International Society of Perinatal Obstetricians, 35(25), 5970–5977. [73] https://doi.org/10.1080/14767058.2021.1903426	USA	Pregnant women hospitalized with moderate COVID-19	748 in total 35 included 17 received remdesivir within 48 h of hospitalization (15/17 with moderate symptoms, 2/17 with critical COVID-19) 7 received remdesivir >48 h of hospitalization 11 did not receive remdesivir	5-day course of remdesivir, antibiotics and/or glucocorticoids	Clinical recovery on day 7 17/17 recovered 7/7 required supplemental oxygen 3/11 recovered Clinical recovery significance between groups: *p* < 0.001
Brooks, K. M., et al. IMPAACT 2032: remdesivir pk & safety in pregnant and non-pregnant women with COVID-19. Topics in Antiviral Medicine; 30(1 SUPPL):267, 2022. [74] https://pesquisa.bvsalud.org/global-literature-on-novel-coronavirus-2019-ncov/resource/en/covidwho-1880041	USA	Pregnant and non-pregnant women with COVID-19	18 in total 10 pregnant (3/10 discontinued early)	200 mg of remdesivir on day 1 followed by 100 mg for 5 to 10 days	Preliminary results were comparable between pregnant and non-pregnant women, but no formal statistical comparisons were made.
Pediatric population					
Ahmed, A., et al. (2022). P168 Remdesivir in the treatment of children 28 days to <18 years of age hospitalised with COVID-19 in the CARAVAN study. Thorax, 77 (Suppl 1), A172–A173. [75] https://doi.org/10.1136/thorax-2022-BTSabstracts.302	USA	Hospitalized children with PCR-confirmed COVID-19	53 76% in need of supplemental oxygen (23% invasive ventilation)	10-day course of IV remdesivir 200 mg on day 1 and 100 mg daily subsequently for children ≥40 kg 5 mg/kg on day 1 and 2.5 mg/kg daily subsequently for children weighing 3 to <40 kg	Clinical outcomes from day 1 to 10 or until discharge and 30-day follow-up On day 10: 75% clinical improvement 60% were discharged By day 30: 85% clinical improvement 83% were discharged A total of 3 deaths documented The incidence of severe adverse events was 21% (none was drug related)
Goldman, D. L., et al. (2021). Compassionate Use of Remdesivir in Children With Severe COVID-19. Pediatrics, 147(5), e2020047803. [76] https://doi.org/10.1542/peds.2020-047803	USA	Hospitalized COVID-19 patients under 18 years old	77	10-day course of IV remdesivir 200 mg on day 1 and 100 mg daily subsequently for children ≥40 kg 5 mg/kg on day 1 and 2.5 mg/kg daily subsequently for children <40 kg	At baseline: 90% required oxygen supplementation 51% required invasive ventilation By 28-day follow-up: 88% had decreased oxygen supplementation needs 83% recovered 73% were discharged Among those in need of invasive ventilation: 90% were extubated 80% recovered 67% were discharged A total of 4 deaths documented There was a low incidence of serious adverse events (16%)
Various populations					
Umemura T., et al. Usage experience of remdesivir for SARS-CoV-2 infection in a patient with chronic cirrhosis of Child–Pugh class C, Journal of Antimicrobial Chemotherapy, Volume 76, Issue 7, July 2021, Pages 1947–1948, [77] https://doi.org/10.1093/jac/dkab076	Japan	Patient with chronic cirrhosis of Child-Pugh class C	1	200 mg first dose and 100 mg per day for 5 days + dexamethasone	On day 4: Fever resolved Respiratory rate normalized On day 12: Oxygen saturation normalized Discharged on day 20 No change found in liver function up to day 39
Ramos-Rincon, J. M., et al. On Behalf Of The Semi-COVID-Network (2022). Remdesivir in Very Old Patients (≥80 Years) Hospitalized with COVID-19: Real World Data from the SEMI-COVID-19 Registry. Journal of clinical medicine, 11(13), 3769. [78] https://doi.org/10.3390/jcm11133769	Spain	Patients ≥ 80 years old with COVID-19	4431 admitted 1312 (30.3%) included (≥80 years old)	140/1312 (10.7%) treated with remdesivir 18/140 treated for 3 days 107/140 treated for 4–5 days 11/140 treated for >5 days	Mortality and 30-day all-cause mortality were lower in remdesivir group vs. control group (*p* < 0.001): Mortality was lower in remdesivir group (OR: 0.45 [95% CI: 0.29–0.69]) 30-day all-cause mortality was lower in remdesivir group (adjusted OR: 0.40 [95% CI: 0.22–0.61])
Wong, C. K. H., et al. (2022). Remdesivir use and risks of acute kidney injury and acute liver injury among patients hospitalised with COVID-19: a self-controlled case series study. Alimentary pharmacology & therapeutics, 56(1), 121–130. [79] https://doi.org/10.1111/apt.16894	China	Patients hospitalized with COVID-19	10.412 in total 860 included	200 mg of remdesivir on day 1 followed by 100 mg for 4 days or until discharge	Risk of ALI, AKI associated with remdesivir: ALI: 334/860 (38.8%) AKI: 137/860 (15.9%) No significantly higher risk with remdesivir use: In the first 2 days: ALI: IRR = 1.261 [95% CI: 0.915–1.737] AKI: IRR = 1.261 [95% CI: 0.899–1.789] Between days 2 and 5: ALI: IRR = 1.087 [95% CI: 0.793–1.489] AKI: IRR = 1.152 [95% CI: 0.821–1.616]
Goldman, J. D., et al. Impact of baseline alanine aminotransferase levels on the safety and efficacy of remdesivir in severe COVID-19 patients. Hepatology; 72(1 SUPPL):279A, 2020. [80] https://pesquisa.bvsalud.org/global-literature-on-novel-coronavirus-2019-ncov/resource/pt/covidwho-986086	USA	Patients with severe COVID-19	397 215 had elevated baseline ALT at initiation of treatment	5- or 10-day course of treatment with remdesivir	AEs and clinical outcomes after treatment: Incidence of serious AEs was similar between groups Hepatobiliary AEs were higher in the high ALT group: 8.8% vs. 3.3% (*p* = 0.068) Symptom duration was longer in the high ALT group (*p* < 0.001) Time to clinical recovery, clinical improvement, and death were similar between groups In severe COVID-19 patients, adverse events and clinical outcomes after RDV initiation were similar among patients with baseline normal ALT and those with elevated ALT

NEWS = National Early Warning Score; eCrCl = estimated Creatinine Clearance; AKI = Acute Kidney Injury; LFTs = Liver Function Tests; CKD = Chronic Kidney Disease; AST = aspartate aminotransferase; ALT = alanine aminotransferase; SRI = Severe Renal Impairment; AEs = Adverse Events; IQR = interquartile range; eGFR = estimated Glomerular Filtration Rate; ALI = Acute Liver Injury.

## Data Availability

Not applicable.

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
