# Peer review of "Remdesivir Use in the Real-World Setting: An Overview of Available Evidence"

_viruses, 2023, doi:10.3390/v15051167_

Round 1

Reviewer 1 Report

This manuscript overviewed the available evidence of real-world data of remdesivir. The outcomes of remdesivir were well introduced, such as safety and effectiveness, and current trends of use. Furthermore, the authors also summarized the effects of combining remdesivir with other therapeutic agents and the situations of using remdesivir on special populations. While there are occasional grammar and style issues, they do not severely affect reader’s understanding. This manuscript is clearly written and is suitable for publication.

Minor issues:

1.       The authors sometimes use “RDV” but sometimes use “remdesivir”, please unify the name.

There are too many to list specifically, but here are some examples:

Page 3, line 92, “remdesivir” is suggested to change to “RDV”;

Page 4, line 121,  “remdesivir” is suggested to change to “RDV”;

Page 4, line 134, “remdesivir” is suggested to change to “RDV”;

Page 4, line 141, “remdesivir” is suggested to change to “RDV”;

Page 5, line 196, “remdesivir” is suggested to change to “RDV”;

Please check the manuscript carefully to revise this issue.

2.       Page 5, line 166, “covid” should be “COVID”;

3.       Page 6, line 222, “2,65” is not clear. Please re-write it to make it clear;

4.       Page 19, line 257, “3,5” should be “3.5”;

5.       Page 19, line 262, “48hours” should be “48 hours”;

6.       Page 19, line 266, “m2” should be “m2”;

7.       Please add a space between the last word and the citation. For example, Pg 19, line 290, “RdRp[87]”, and line 292, “coronaviruses[88]”;

8.       Page 20, line 315, please remove one dot from the title of “7. . Limitations”.

Reviewer 2 Report

Manuscript “Remdesivir Use in the Real-World Setting: an Overview of Available Evidence” is informative manuscript. The manuscript needs revision before publication. I have the following comments/suggestions for authors to address.

1.     Check the abbreviations throughout the manuscript and introduce the abbreviation when the full word appears the first time in the abstract and the remaining for the text and then use only the abbreviation (For example, ICU, FDA). Make a word abbreviated in the article that is repeated at least two times in the text, not all words to be abbreviated (For example, MERS-CoV).

2.     “MERS-Co-V”---“MERS-CoV”

3.     There is a lack of recent literature citations. For example, in lines 51-54 “Remdesivir (RDV) branded under the name Veklury is an antiviral agent that has previously demonstrated antiviral activity against filoviruses (Ebola viruses, Marburg virus), coronaviruses (SARS-CoV, MERS-Co-V, SARS-CoV-2), paramyxoviruses (parainfluenza type III virus, Nipah virus, Hendra virus, measles, and mumps virus), and Pne-54 moviridae (respiratory syncytial virus) (DOI: 10.1002/jmv.27517)”; 89-91 “RDV is a phosphoramidate prodrug of a monophosphate nucleoside and acts as a viral RNA-dependent RNA polymerase (RdRp) inhibitor, targeting the viral genome replication process. (DOI: 10.3390/biomedicines9060689; DOI: 10.1016/j.ejmech.2020.112527)”.

4.     The conclusion seems very general, lacking the future aspects and major findings. Authors have suggested to provide additional insights or recommendations that could strengthen the study's findings and contribute to the field's understanding of Remdesivir's role in managing COVID-19. For example, “Remdesivir, the intravenously administered version (version 1.0) of GS-441524, is the first FDA-approved agent for SARS-CoV-2 treatment. However, clinical trials have presented conflicting evidence on the value of remdesivir in COVID-19. Therefore, oral GS-441524 derivatives (VV116 and ATV006; version 2.0, targeting highly conserved viral RdRp) could be considered as game-changers in treating COVID-19 because oral administration has the potential to maximize clinical benefits, including decreased duration of COVID-19 and reduced post-acute sequelae of SARS-CoV-2 infection, as well as limited side effects such as hepatic accumulation.” (DOI: 10.3389/fimmu.2022.1015355).

5.     In conclusion and future perspectives section, it is better to include a paragraph describing the perspective of what are the desired properties for the next-generation of anti-SARS-CoV-2 drugs?

Minor editing of English language required
